# TR-Rules: Rule-based Model for Link Forecasting on Temporal Knowledge Graph Considering Temporal Redundancy

**Ningyuan Li[1], Haihong E[1]***, **Li Shi[2], Mingzhi Sun[1], Tianyu Yao[1],**
**Meina Song[1]**, **Yong Wang[3]**, **Haoran Luo[1]**

[1]Beijing University of Posts and Telecommunications;
[2]National Computer Network Emergency Response Technical Team/Coordination
Center of China; [3]Lianyang Guorong (Beijing) Technology Co., Ltd.
{jason.ningyuan.li, ehaihong, sunmingzhi, yaotianyu, mnsong, luohaoran}@bupt.edu.cn
shili@cert.org.cn, wangyong@oceanum.cn

## Abstract

Temporal knowledge graph (TKG) has been proved to be an effective way for modeling dynamic facts in real world. Many efforts have been devoted into predicting future events i.e. extrapolation, on TKGs. Recently, rule-based knowledge graph completion methods which are considered to be more interpretable than embedding-based methods, have been transferred to temporal knowledge graph extrapolation. However, rule-based models suffer from temporal redundancy when leveraged under dynamic settings, which results in inaccurate rule confidence calculation. In this paper, we define the problem of temporal redundancy and propose TR-Rules which solves the temporal redundancy issues through a simple but effective strategy. Besides, to capture more information lurking in TKGs, apart from cyclic rules, TR-Rules also mines and properly leverages acyclic rules, which has not been explored by existing models. Experimental results on three benchmarks show that TR-Rules achieves state-of-the-art performance. Ablation study shows the impact of temporal redundancy and demonstrates the performance of acyclic rules is much more promising due to its higher sensitivity to the number of sampled walks during learning stage. [1]

## 1 Introduction

Knowledge Graphs (KGs) store various real-world facts in the form of (s,r,o) in which s and o represent subject and object real-world entities while r denotes a relation between s and o. KGs are critical in many downstream applications such as question answering (Hao et al., 2017), recommender systems (Hildebrandt et al., 2019) and information retrieval (Liu et al., 2018). Traditional knowledge graph can be viewed as a static knowledge base, however, plausibility of most real-world facts are

---

*Corresponding Author
[1]Our code is available at https://github.com/JasonLee-22/TR-Rules

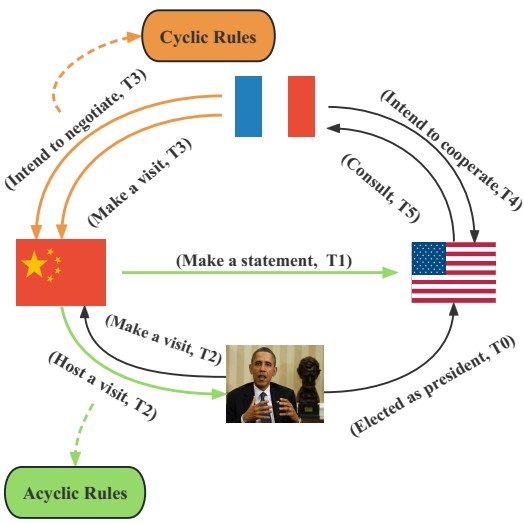

Figure 1: An example subgraph of Temporal knowledge graph involving political facts and there are cyclic and acyclic rules existing in it which provide crucial information for link forecasting.

dependent on time, i.e., facts in KGs may not stand perpetually. Thus, TKGs are introduced to resolve this limitation, where each fact is represented by a quadruple (s,r,o,t) with timestamp t incorporated.

Link forecasting or extrapolation on TKG which aims at predicting possible links at future timestamps, has been one of the main tasks being studied. Recently, embedding-based models like RE-NET(Jin et al., 2019), CyGNet(Zhu et al., 2021), have been proposed and perform well for extrapolation on TKGs. However, compared with rule-based models, embedding-based models are believed lacking interpretability which leads to limitations on their downstream applications. TLogic(Liu et al., 2022) is the first model which leverages rule-based methods on TKG link forecasting and gains competitive results as well as more explainable predicting process since it generates human readable rules. It mines cyclic rules by performing temporal random walk on TKGs first, and then extracting and generalizing walks into symbolic rules.

Nevertheless, existing models only mine cyclic rules in TKGs and ignore acyclic rules which, also contain important information for link prediction. Besides, previous works do not notice that under temporal settings, (s,r,o) which has to be unique in static KGs, might appears repeatedly with different timestamps. And this temporal property can cause the model to obtain inappropriate confidence of rules through conventional confidence calculation algorithm, which further leads to incorrect predictions. We call this issue Temporal Redundancy.

In this paper, we first define Temporal Redundancy and give an intuitive illustration on why Temporal Redundancy causes inappropriate confidence calculation. Then we propose TR-Rules, a rule-based TKG link forecasting model which resolves Temporal Redundancy through a simple but effective aggregation strategy. TR-Rules aggregates those matched rule bodies before any new rule heads appear as one body during the rule support counting process. Unlike previous works merely mine cyclic rules on TKG, by performing temporal random walk on TKG, TR-Rules also mines acyclic rules which is crucial for link prediction but unexplored by existing models. Experimental results show that, without considering temporal redundancy, simply using acyclic rules together with cyclic rules gives worse performance, which is the probable reason why acyclic rules have not been involved and studied by existing models. Ablation study also demonstrates that acyclic rules are more sensitive to the number of random walks sampled for rule mining, which consequently results in a more promising performance.

Our contributions are as follows:

- We discover and define the problem of temporal redundancy which results in inappropriate rule confidence calculation.

- We propose TR-Rules, a rule-based temporal knowledge graph extrapolation model which resolves temporal redundancy with a simple algorithm and, to our best knowledge, is the first model that mines acyclic rules on temporal knowledge graphs.

- We evaluate TR-Rules on three benchmarks and the results show that TR-Rules achieves state-of-the-art performance.

- We also study the performance of acyclic rules and cyclic rules of different length respec-

tively and experimental results prove the effectiveness of TR-Rules in solving temporal redundancy and demonstrate that the performance of acyclic rules is more promising due to its higher sensitivity to the number of sampled walks.

## 2 Related Work

### 2.1 TKG Extrapolation

Most of the existing TKG extrapolation models are embedding-based models which aim at learning representations of entities, relations as well as timestamps in vector space and then obtain scores of facts through proposed score functions to measure the plausibility of those facts being true. RE-NET (Jin et al., 2019) employs RNN-based encoder and RGCN-based encoder to capture sequential and structural information of facts for future events prediction. CyGNet (Zhu et al., 2021) introduces copy-generation mechanism to learn more information from repetitive facts in history. HIP (He et al., 2021) proposes three score functions to pass information from temporal, structural and repetitive patterns. CluSTeR (Li et al., 2021) adopts reinforcement learning to search clues from history for link forecasting. CENET (Xu et al., 2022) focuses more on the non-historical facts and uses contrastive learning to distinguish whether historical or non-historical information is more important for events prediction.

Apart from embedding-based models, xERTE (Han et al., 2021) and TLogic (Liu et al., 2022) provide more explainable predictions. xERTE performs extraction on subgraphs around the queries to model structural dependencies and temporal dynamics. TLogic mines temporal rules via adopting temporal random walks on TKGs, which are applied to forecast plausible events at future timestamps afterwards.

### 2.2 Rule-based KG Completion

AMIE (Galárraga et al., 2013) proposes one of the earliest rule mining systems that learns closed rules on knowledge bases. AnyBURL (Meilicke et al., 2019) is a random-walk-based model which transforms sampled walks from KG into general rules and then applies obtained rules to predict missing facts. SAFRAN (Ott et al., 2021) focuses on the effect of rules redundancy on aggregation especially noisy-or aggregation, and proposes an approach to cluster rules before application.

## 3 Preliminaries and Notations

### 3.1 TKG Extrapolation

In this paper, we use $\mathcal{E}$, $\mathcal{R}$ and $\mathcal{T}$ to represent all the entities, relations and timestamps respectively. While $|\mathcal{E}|$ and $|\mathcal{R}|$ denote the number of entities and relations. TKG can be considered as a sequence of graphs i.e. $\mathcal{G} = \{G_1...G_n\}$ in which $G_i = \{(s, r, o, t_i)\}$ where $s, o \in \mathcal{E}$, $r \in \mathcal{R}$ and $t_i \in \mathcal{T}$. TKG extrapolation is to answer a query $(s, r, ?, t_q)$ or $(?, r, o, t_q)$ based on the previously observed facts $\{G_i | t_i < t_q\}$. Normally, the model generates a ranked list of candidate entities according to their plausibility.

### 3.2 Random Walk on TKG

Following the definition given by TLogic, a non-increasing random walk on TKG with length $l \in \mathbb{N}$ can be defined as:

$$((e_{l+1}, r_l, e_l, t_l), (e_l, r_l, e_{l-1}, t_{l-1}), ..., (e_2, r_1, e_1, t_1))$$
$$with \quad t_l \geq t_{l-1}... \geq t_1$$

The non-increasing restriction ensures random walks only go back in time forming chains complying with temporal causality.

### 3.3 Temporal Rules

Here, we give the definition of temporal cyclic and acyclic rules. Let $E_i$ and $T_i$ be variables and $r_i, e_i$ represent fixed constants (fixed constant means that it has to be a specific entity or relation). A temporal cyclic rule can be represented as:

$$(E_1, r_h, E_{n+1}, T_{n+1}) \leftarrow \bigwedge_i^n (E_i, r_{bi}, E_{i+1}, T_i)$$

where $T_{n+1} > T_n \geq ... \geq T_1$.

And a temporal acyclic rule can be defined as:

$$(E_a, r_h, e_h, T_h) \leftarrow (E_a, r_b, e_b, T_b)$$

where $T_h > T_b$, $e_h$ and $e_b$ are fixed entities. It is notable that in TR-Rules, just as the definition given above, we confine the length of acyclic rules to be **1**. The left side of a rule is called rule head and the right side of a rule is called rule body. A temporal rule in the above forms indicates that when the rule body stands(satisfying the variable constraints and temporal restrictions), the rule head at future timestamp $T_{n+1}$ will be true. When applied, each variable in rule head or rule body can be instantiated with specific entities or relations.

For better illustration, we give two examples of cyclic and acyclic rules, both of which are reasonable and critical in link forecasting. For facts of $(Ameriaca, Intent\ to\ negotiate, China, T_1)$ and $(America, Make\ a\ visit, China, T_2)$ where $T_1 < T_2$, we can generalize them into a cyclic rules of length 1:

$$(X, Make\ a\ visit, Y, T_h) \leftarrow$$
$$(X, Intent\ to\ negotiate, Y, T_b)$$

From facts $(Jason, Major, Education, T_1)$ and $(Jason, Job, Teacher, T_2)$, a reasonable temporal acyclic rule can be generalized which is:

$$(X, Job, Teacher, T_h) \leftarrow$$
$$(X, Major, Education, T_b)$$

### 3.4 Confidence Estimation

Generally, temporal rules may not always be a tautology. Thus, we need to estimate the confidence of each temporal rule before application. Existing rule-based TKG extrapolation model TLogic leverages the standard confidence which is widely used in static KG completion models and take timestamps into consideration. Take a cyclic rule R: $(E_1, r_h, E_{l+1}, T_{l+1}) \leftarrow \bigwedge_i^l (E_i, r_{bi}, E_{i+1}, T_i)$ for example. The body support of R can be defined as the number of rule body instances, i.e., the number of sequences $((e_1, r_{b1}, e_2, t_1), ..., (e_l, r_{bl}, e_{l+1}, t_l))$ where $(e_i, r_{bi}, e_{i+1}, t_i) \in \mathcal{G}$ and $t_i \leq t_{i+1}$ for $i \in [0, l-1]$. And the rule support of R is defined as the number of body instances whose rule head stands at future timestamp $t_{l+1}$, i.e., $(e_1, r_h, e_{l+1}, t_{l+1}) \in \mathcal{G}$. The confidence is calculated as

$$conf = w \cdot \frac{rule\_sup}{body\_sup + c} \qquad (1)$$

where $c$ is a hyper-parameter for smoothing and $w$ is a weight for rules of different types and different lengths. This confidence estimation algorithm works for static KG completion but when it comes to temporal settings, it suffers from Temporal Redundancy which will be defined and discussed in the next section.

## 4 Proposed Method

In this section, we propose TR-Rules, a rule-based model for TKG extrapolation, which adopts both cyclic and acyclic rules mining and resolves temporal redundancy. First, we give the definition of **Temporal Redundancy** and illustrate how TR-Rules

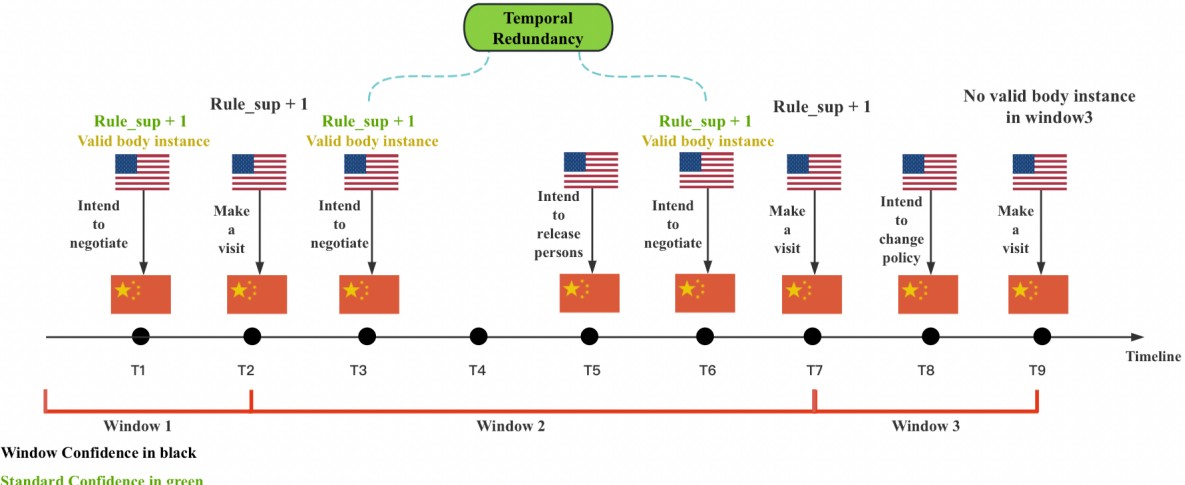

Figure 2: Illustration of the issue of Temporal Redundancy and window confidence.

resolves it. Then we introduce the rule learning module of TR-Rules where specifically we show the acyclc rules mining algorithm of TR-Rules. Finally, the rule application module of TR-Rules is introduced to show how we combine cyclic and acyclic rules together to predict future events.

### 4.1 Temporal Redundancy

As mentioned above, standard confidence counts the number of body instances whose rule heads stand at future timestamps. It works well for static KGs, however, it is notable that in TKGs a triplet can appear more than once, which means that it is possible the same body instance can be counted multiple times for rule support only if it holds true at different timestamps before the timestamp where its corresponding rule head stands. Especially, when there are multiple body instances holding true between any two temporally adjacent rule head instances, we call this phenomenon Temporal Redundancy.

In figure 2, Temporal Redundancy takes place between $T_2$ and $T_7$. When Temporal Redundancy happens, the confidence of rules is over estimated, because many interactions between facts have strongly connections with the accumulation of the causes. In figure 2, we take a cyclic rule: $(X, Make \ a \ visit, Y) \leftarrow (X, Intent \ to \ negotiate, Y)$ which is mined from ICEWS14 and some relevant facts for example. In this case, when estimating the confidence of this rule, suppose there is no other relevant facts, according to the defini-

tion of standard confidence, the result should be 1. Because there are 3 body instances $(The \ US \ envoy, Intent \ to \ negotiate, China)$ at $(T_1, T_3, T_6)$, which means the body support is 3 and for body instance at $T1$, there are head instances $Intent \ to \ negotiate$ holding true at $(T_2, T_9)$ making it counted as a rule support, for body instances at $(T_3, T_6)$, head instance at $T_9$ makes them counted as rule support. Thus the rule support is 3 either and ignoring the parameters for smoothing, the confidence of this rule is 1. However, in fact, the head instance $(The \ US \ envoy, Make \ a \ visit, China)$ standing at $T_7$ might be the result of the two intentions at $(T_3, T_6)$ together. In other word, only a single intention may not necessarily results in a visit, which intuitively contradicts to the confidence being 1. Since when the confidence of a rule being 1, it means that the rule head will surely hold true at future timestamps as long as there exists valid body instances. From this point of view, standard confidence calculation is not suitable for temporal settings.

In TR-Rules, we solve this issue by proposing a simple but effective window based algorithm for confidence estimation called window confidence. Instead of focusing on counting the number of body instances whose rule heads stand in future as the rule support, window confidence calculates the rule support based on the rule head instances. During the confidence estimation of each rule, we sort the timestamps of all the rule head instances and

divide the timeline into windows by those timestamps. Each window starts at timestamp 0 or a timestamp of rule head instances mentioned above and ends at the timestamp next to its start point or the last timestamp. Like in figure 2, we obtain 3 windows according to the rule head instances. Starting from the first window, if there exists at least one body instance, then we add 1 to the rule support and shift to the next window. We can see that by using window confidence, the confidence of the example rule above is $\frac{1}{2}$ since only window 1 and window 2 involve valid body instances. What window confidence intends to do is that aggregate the temporal redundancy inside windows with a designed function. In TR-Rules, the function we utilize just aggregate all body instances inside a window as one. From this point of view, the standard confidence applied in TLogic is a trivial version of window confidence which associates with no aggregation function.

## 4.2 Rule Learning

In this section we introduce the learning procedures of cyclic and acyclic rules respectively. Our cyclic rules mining algorithm is similar to the rule learning stage in TLogic. We first randomly sample a quadruples $(e_1, r_h, e_{n+1}, t_{n+1})$ which is considered to be the rule head. Then we perform random walks traversing back in time and starting from the entity $e_{n+1}$. For the first step, to satisfy the temporal restriction mentioned in section 3.3, the random walker is confined to sample edges connecting to the current node at timestamps $t_n$ where $t_{n+1} > t_n$. Then, in the following steps, the walker samples adjacent edges whose timestamps $t_i$ satisfies $t_i \leq t_{i+1}$ where $2 \leq i \leq n-1$ iteratively. When it comes to the last step, the walker is restricted to sample a valid edge that connects to $e_1$ so that all the walks together with the rule head form a cyclic path. If there is no available edges during any steps, the current random walk will stop and we will start a new one by randomly sampling a rule head. Since we traverse back in time during the learning procedure, when generating temporal rules, we need to rebuild the path in reverse order by substituting relations in edges with their inverse and exchanging the objects and subjects which gives: $(e_1, r'_1, e_2, t_1)...(e_n, r'_n, e_{n+1}, t_{n+1})$. Finally, we replace the entities and timestamps with variables and the generated rules with length $n$ are specific to relation $r_h$. Specifically, if an entity

appears multiple times in a rule, then it has to be replaced by the same variable.

In TR-Rules, when sampling the new edge from current node, we use the exponentially weighted transition distribution which can be defined as:

$$\mathbf{P}(l; t_c, C) = \frac{exp(t_l - t_c)}{\sum_{t_{l'} \in \mathcal{T}_c} exp(t_{l'} - t_c)}$$

where $C, t_c$ represent the current edge and the timestamp of current edge and $\mathcal{T}_c$ denotes a set of possible timestamps associated with available edges in the next step:

$$\begin{cases} \{t | (e_{n+1}, r, e, t) \in G \text{ and } t < t_c\} \text{ step } 1, \\ \{t | (obj_c, r, e_1, t) \in G \text{ and } t \leq t_c\} \text{ step } n, \\ \{t | (obj_c, r, e, t) \in G \text{ and } t \leq t_c\} \text{ otherwise.} \end{cases}$$
(2)

where $obj_c$ represents the object entity of the current edge, $e \in \mathcal{E}$ and $G$ denotes the whole TKG. This distribution gives higher probability when $t_l$ is close to $t_c$ which makes the random walker more likely to select closer edges during sampling. This is based on the intuition that closer facts might contribute more to predictions compared with facts that happen at further timestamps.

In TR-Rules, we also propose a learning algorithm for acyclic rules. Similarly, we first sample a random quadruples $(e_h, r_h, e_i, t_h)$ to be the rule head. Then, unlike the cyclic mining procedure, starting from $e_h$, the random walkers are made to select edge adjacent to $e_h$ which is $(e_h, r_b, e_j, t_b)$ where $t_b < t_h$. Then we replace $e_h$ with a variable and set $e_i, e_j$ to be fixed. Consequently, an acyclic rule specific to $r_h$ is obtained.

When estimating confidence, we first sample a fixed number of instanced paths and leverage the window confidence mentioned above. The output of the rule learning module is a set of rules with their corresponding confidence.

## 4.3 Rule Application

Given a query $Q : (s, r, ?, t)$, we first find the rules whose head relations match with $r$ in $Q$ and sort them in descending order according to their confidence. Then, for each rule, we traverse the TKGs to find matching body instances which satisfy the temporal constraints and the object entity in each rule head is considered to be one of the answer candidates if there exist valid body instances. For a candidate answer $c$ generated by rule $R$, we leverage the score function $f$ proposed in TLogic to

measure the plausibility of $(s, r, c, t)$ being true:

$$f(R, c) = a \cdot conf(R)+ \\ (1 - a) \cdot exp(-\lambda(t - t_1(\mathcal{B}(R, c)))) \quad (3)$$

where $a$, $\lambda$ are hyper-parameters, $\mathcal{B}(R, c)$ denotes all the body instances of rule $R$ that leads to candidate $c$ and $t_1(b)$ means the earliest timestamp in the body instance $b$. If there are multiple body instances, then $t_1$ returns the one closest to $t$. This score function considers not only the rule confidence but also the time difference. Obviously, the confidence part enables candidates derived by rules with higher confidence to receive higher scores. And the time difference part assigns higher scores to candidates generated by close body instances, which is based on the phenomenon that edges in rules incline to hold true when the time difference is low. Normally, one candidate can be generated by multiple rules, we use Noisy-OR to aggregate scores of $c$ obtained by different rules. The final scores of $c$ is calculated as:

$$score(c) = 1 - \prod_{s \in \mathcal{C}}(1 - s) \quad (4)$$

where $\mathcal{C}$ represents all the scores of candidate $c$ derived from multiple rules. The intuition behind Noisy-OR is that the results represent the possibility that at least one rule holds for candidate $c$.

It is possible that no rule learned can answer the given query. In this paper, we follow the simple baseline proposed in TLogic which will generate candidate answers according to the object distribution in the training set.

## 5 Experiments

### 5.1 Experimental Setup

**Datasets** We evaluate TR-Rules on three datasets: ICEWS14, ICEWS18 and ICEWS0515. All of the three datasets are subsets of Integrated Crisis Early Warning System (Boschee et al., 2015), which store political events take place in 2014, 2018 and from 2005 to 2015 respectively. Under TKG extrapolation settings, all datasets are split into train, valid and test according to the timestamps in ascending order, which do not overlap with each other. The statistics of these three datasets are listed in Table 1.

**Metrics and Implementation Details** The metrics we use to evaluate TR-Rules are: mean reciprocal rank (MRR) and Hits@1/3/10. For N testing queries, the MRR is computed as $MRR = \frac{1}{N} \sum_i \frac{1}{rank_i}$ and Hits@K is calculated as $Hits@K = \frac{1}{N} \sum_i \mathbf{I}(rank_i < K)$, where $\mathbf{I}$ is the indicator function. In this paper, we use the time-aware filtering protocol proposed in xERTE(Han et al., 2021), which filters out all the true entities $\{o | (s, r, o, t) \in G_t\}$ given query $q : (s, r, ?, t)$, except the answer entity of query $q$. Compared with the traditional filtering protocol that filters out all the true entities $o$ as long as the triplet $(s, r, o)$ holds true at any timestamps, time-aware filtering is more reasonable.

As for the implementation details, we learn cyclic rules with length of 1,2,3 and acyclic rules. We set the the number of sampled walks during rules learning stage to 200 for cyclic rules on all datasets and for acyclic rules, we set it to 5000 on ICEWS14 and 1000 on ICEWS0515 and ICEWS18. During the confidence estimation process, we sample 500 body instances for all kinds of rules. The smoothing parameter $c$ is 3 and we filter out rules with confidence less than 0.01 and body instances number less than 2. We set $w$ in (1) to 1 for cyclic rules in all lengths and 0.5 for acyclic rules. $a, \lambda$ in (3) are set to 0.5 and 1 for all three datasets. Following TLogic, we also set time windows during rule applications which filter out facts not in the period spanning from $t_q - w_{size}$ to $t_q$, where $t_q$ represents the timestamp of the query and $w_{size}$ denotes the time window size. We $w_{size}$ to 200 for ICEWS18, 1000 for ICEWS0515 and use all facts on ICEWS14, which is the same as TLogic does.

**Baseline Methods** In this paper, we select both static and temporal state-of-the-art models as baselines. As for static models we select Dismult(Yang et al., 2014), ComplEx(Trouillon et al., 2016) and AnyBURL(Meilicke et al., 2019). Temporal models include TTransE (Leblay and Chekol, 2018), TA- DistMult (García-Durán et al., 2018), DE-SimplE(Goel et al., 2020), TNTComplEx (Lacroix et al., 2020), RE-Net (Jin et al., 2019), CyGNet (Zhu et al., 2021), xERTE (Han et al., 2021),

| | $N_{train}$ | $N_{valid}$ | $N_{test}$ | $N_{ent}$ | $N_{rel}$ | $N_{time}$ |
|---|---|---|---|---|---|---|
| ICEWS14 | 63685 | 13823 | 13222 | 7128 | 230 | 365 |
| ICEWS0515 | 322958 | 69224 | 69147 | 10488 | 251 | 4017 |
| ICEWS18 | 373018 | 45995 | 49545 | 23033 | 256 | 304 |

Table 1: Statistics of ICEWS14, ICEWS05-15 and ICEWS18

| | ICEWS14 | | | | ICEWS05-15 | | | | ICEWS18 | | | |
|---|---|---|---|---|---|---|---|---|---|---|---|---|
| Model | MRR | Hits@1 | Hits@3 | Hits@10 | MRR | Hits@1 | Hits@3 | Hits@10 | MRR | Hits@1 | Hits@3 | Hits@10 |
| Dismult | 27.67 | 18.16 | 31.15 | 46.96 | 28.73 | 19.33 | 32.19 | 47.54 | 10.17 | 4.52 | 10.33 | 21.25 |
| ComplEx | 30.84 | 21.51 | 34.48 | 49.58 | 31.69 | 21.44 | 35.74 | 52.04 | 21.01 | 11.87 | 23.47 | 39.87 |
| AnyBURL | 29.67 | 21.26 | 33.33 | 46.73 | 32.05 | 23.72 | 35.45 | 50.46 | 22.77 | 15.10 | 25.44 | 38.91 |
| TTransE | 13.43 | 3.11 | 17.32 | 34.55 | 15.71 | 5.00 | 19.72 | 38.02 | 8.31 | 1.92 | 8.56 | 21.89 |
| TA-Dismult | 26.47 | 17.09 | 30.22 | 45.41 | 24.31 | 14.58 | 27.92 | 44.21 | 16.75 | 8.61 | 18.41 | 33.59 |
| DE-SimplE | 32.67 | 24.43 | 35.69 | 49.11 | 35.02 | 25.91 | 38.99 | 52.75 | 19.30 | 11.53 | 21.86 | 34.80 |
| TNTComplEx | 32.12 | 23.35 | 36.03 | 49.13 | 27.54 | 19.52 | 30.80 | 42.86 | 21.23 | 13.28 | 24.02 | 36.91 |
| RE-Net | 38.28 | 28.68 | 41.34 | 54.52 | 42.97 | 31.26 | 46.85 | 63.47 | 28.81 | 19.05 | 32.44 | 47.51 |
| CyGNet | 32.73 | 23.69 | 36.31 | 50.67 | 34.97 | 25.67 | 39.09 | 52.94 | 24.93 | 15.90 | 28.28 | 42.61 |
| CENET | - | - | - | - | 37.16 | 27.78 | 41.16 | 55.49 | 27.14 | 18.58 | 29.99 | 44.15 |
| xERTE | 40.79 | 32.70 | 45.67 | 57.30 | 46.62 | **37.84** | 52.31 | 63.92 | 29.31 | 21.03 | 33.51 | 46.48 |
| TLogic | 43.04 | 33.56 | 48.27 | **61.23** | 46.97 | 36.21 | 53.13 | 67.43 | 29.82 | 20.54 | 33.95 | 48.53 |
| **TR-Rules** | **43.32** | **33.96** | **48.55** | 61.17 | **47.64** | 37.06 | **53.80** | **67.57** | **30.41** | **21.10** | **34.58** | **48.92** |

Table 2: Results of TR-Rules on ICEWS14, ICEWS05-15 and ICEWS18. Best results are in bold and second best results are underlined.

TLogic(Liu et al., 2022) and CE-NET(Xu et al., 2022). All the results except CENET are taken from (Liu et al., 2022) and we evaluate CENET on ICEWS0515 and ICEWS18 using time-aware filtering protocol.

## 5.2 Main Results

Table 2 reports results of TR-Rules and all baseline models under time-aware filtering protocol on three datasets. We can see that TR-Rules achieves state-of-the-art performance in all metrics except Hits@10 on ICEWS14 and Hits@1 on ICEWS0515 where TR-Rules gives the second best results. Table 3 presents some high confidence rules of variant types mined by TR-Rules, which demonstrates the better interpretability of TR-Rules.

## 5.3 Ablation Study

To study the contribution of acyclic rules and the impact of Temporal Redundancy on different kinds of rules, we perform ablation study on three datasets and the results are given in Table 4. As for the notations, C1 means we only mine cyclic rules of length one, A means we only mine acyclic rules, C means we only mine cyclic rules but including length of 1,2,3 and R means we use window confidence for solving Temporal Redundancy. TR-Rules(C) is equivalent to TLogic which mines cyclic rules in TKGs.

We can see that the introduction of window confidence that addresses Temporal Redundancy improves the performance of TR-Rules in most cases including mining only cyclic rules or only acyclic rules or both. Besides, window confidence improves the performance of acyclic rules more than that of cyclic rules on three datasets. The possi-

ble explanation is that, first, we mine more acyclic rules, the superiority of window confidence is amplified. In addition, without the cyclic restriction, the matched body instances of acyclic rules are slightly less likely to be relevant to the corresponding head instances. In other words, acyclic rules suffer more from temporal redundancy due to its acyclic forms. However, the aggregation operation in window confidence perfectly alleviate this issue. This also explains why TR-Rules(C+A) which just additionally involves acyclic rules comparing to TLogic, gives worse results than TLogic does. In other words, under temporal settings, window confidence provides the proper way to make use of acyclic rules through fixing the bias in confidence calculation caused by temporal redundancy. As discussed in TLogic, we can see that only cyclic rules of length 1 can give competitive performance on all datasets.

## 5.4 Acyclic Rules Performance Analysis

We also study the performance of both kinds of rules on ICEWS14 when the number of sampled walks during rule learning are set to different values. In figure 3, the orange line displays the performance of acyclic rules with standard confidence being utilized, the blue line is the performance of acyclic rules with window confidence being utilized and the green line is the performance of cyclic rules with standard confidence. As we can see in figure 3, the performance of acyclic rules with whichever confdence calculation algorithms improves dramatically as the number of sampled walks grows. The improvements are up to **14.89%(Hits@10)** and **10.07%(MRR)**. However, in comparison, the performance of cyclic rules changes no more than 0.2% as the number of walks

| Rules | Confidence | Type |
|---|---|---|
| $(X, Engage\ in\ negotiations, Y, T_h) \leftarrow (X, Express\ intent\ to\ ease\ sanction, Y, T_b)$ | 0.80 | Cyclic1 |
| $(X, Conduct\ strike, Y, T_h) \leftarrow (X, Break\ diplomatic\ relations, Y, T_{b1}) \wedge$ $(Y, Expel\ peacekeepers^{-1}, X, T_{b2}) \wedge (X, Break\ diplomatic\ relations, Y, T_{b3})$ | 0.82 | Cyclic3 |
| $(X, Reduce\ relation, South\ Korea, T_h) \leftarrow (X, Express\ intent\ to\ ease\ sanction, North\ Korea, T_b)$ | 0.86 | Acyclic |

Table 3: High confidence rules mined by TR-Rules

| Model | ICEWS14 | | | | ICEWS05-15 | | | | ICEWS18 | | | |
|---|---|---|---|---|---|---|---|---|---|---|---|---|
| | MRR | Hits@1 | Hits@3 | Hits@10 | MRR | Hits@1 | Hits@3 | Hits@10 | MRR | Hits@1 | Hits@3 | Hits@10 |
| TLogic | 43.04 | 33.56 | 48.27 | **61.23** | 46.97 | 36.21 | 53.13 | 67.43 | 29.82 | 20.54 | 33.95 | 48.53 |
| TR-Rules(C1) | 40.57 | 30.86 | 46.32 | 59.16 | 44.98 | 34.18 | 51.42 | 65.57 | 27.73 | 18.65 | 31.70 | 46.64 |
| TR-Rules(C1+R) | 40.79 | 31.16 | 46.57 | 59.21 | 45.76 | 35.07 | 52.26 | 66.10 | 27.83 | 18.74 | 31.90 | 46.68 |
| TR-Rules(A) | 31.14 | 24.60 | 35.11 | 43.75 | 30.28 | 23.88 | 34.21 | 42.72 | 17.02 | 11.86 | 19.51 | 27.85 |
| TR-Rules(A+R) | 31.98 | 25.33 | 35.97 | 44.83 | 30.85 | 24.56 | 34.57 | 43.11 | 17.28 | 12.11 | 19.81 | 28.21 |
| TR-Rules(C+R) | 43.03 | 33.54 | 48.33 | 61.07 | **47.71** | **37.10** | **53.91** | **67.72** | 29.87 | 20.55 | 33.94 | 48.63 |
| TR-Rules(C+A) | 41.58 | 32.54 | 46.56 | 59.23 | 46.08 | 35.44 | 52.24 | 66.25 | 29.56 | 20.36 | 33.68 | 48.12 |
| **TR-Rules** | **43.32** | **33.96** | **48.55** | 61.17 | 47.64 | 37.06 | 53.80 | 67.57 | **30.41** | **21.10** | **34.58** | **48.92** |

Table 4: Ablation study of TR-Rules on ICEWS14, ICEWS05-15 and ICEWS18. Best results are in bold and second best results are underlined.

grows. The possible reason is that acyclic rules involve fixed entities (e.g. South Korea instead of Country) while entities are replaced with variables in cyclic rules. Hence, acyclic rules need to capture enough information in TKGs by more samples and cyclic rules need less sampled walks because only some specific walks can be generalized into a paradigm. The performance of cyclic rules even get slightly worse because 200 sampled walks are enough for mining strong and general rules and as sampling more walks, more rare but accidental rules are obtained which might affect the inference accuracy. Thus, we can conclude that acyclic rules are more sensitive to the number of walks and yield promising results.

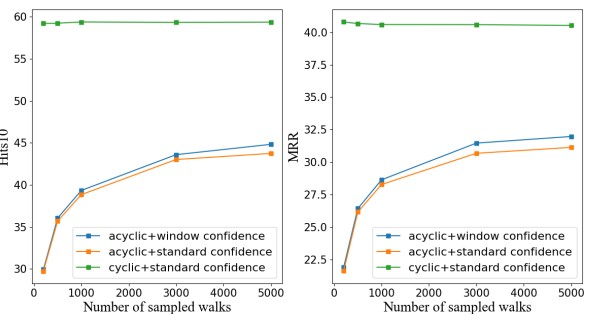

Figure 3: Results of cyclic rules and acyclic rules with different number of sampled walks on ICEWS14.

## 5.5 Temporal Redundancy Impact Analysis

As we have discussed in section 5.3, window confidence benefits the performance of both cyclic rules and acyclic rules. In figure 3, we can clearly see the difference between the orange line and blue

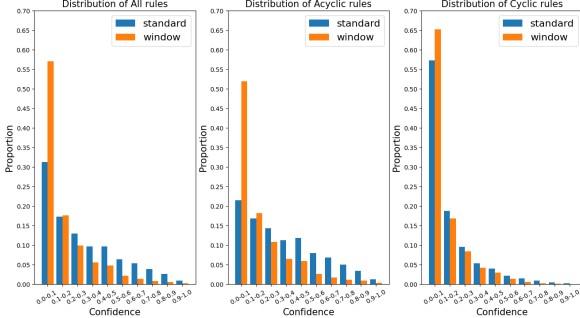

Figure 4: The confidence distribution of rules mined by TR-Rules(window) and TLogic(standard) on ICEWS0515.

line increases which means the impact of Temporal Redundancy also grows as the number of sampled walks gets larger and window confidence resolves it effectively. Figure 4 describes the distribution of confidence of rules calculated by standard and window confidence algorithm. As we can see in the figure on the left which displays the confidence distribution of all rules, generally, confidence computed by window confidence algorithm inclines to be in lower confidence section. This phenomenon coincides with the intuition that by solving temporal redundancy, TR-Rules gives lower but more practical confidence under temporal settings, which can be viewed as refinements for temporal rules and the original inflated confidence is corrected. It demonstrates that our method indeed works and verifies the correctness of our motivations. The rest two figures display the confidence distribution of acyclic rules and cyclic rules respectively. We can see that the difference between these two distribu-

tions is larger in the middle figure which supports our conclusion of acyclic rules suffer more from temporal redundancy. It further demonstrates that our proposed window confidence serves as an effective way to utilize acyclic rules in TKGs for extrapolation task.

## 6 Conclusion

In this paper, we define the problem of Temporal Redundancy and propose TR-Rules, a rule-based TKG extrapolation model which solves Temporal Redundancy by replacing standard confidence with window confidence. TR-Rules also firstly mines acyclic rules in TKGs which are proved to have more promising performance compared with cyclic rules because of their higher sensitivity to the number of sampled random walks. Experimental results show that TR-Rules achieves state-of-the-art performance in most metrics on three datasets. As for future work, it is promising to explore more sophisticated aggregation methods for window confidence calculation, such as employing cluster or machine learning methods.

## Limitations

As we discussed in Section 4.1, the aggregation function in window confidence is quite simple. It cannot model the interactions among body instances in different windows which might result inaccurate confidence in some cases. Besides, for rule-based models, especially those random-walk-based models like TR-Rules and TLogic, it is impossible to mine very long rules due to the unaffordable time consumption. In TR-Rules, we mine cyclic rules of length 1,2,3 and acyclic rules of length 1. Thus, as we discuss in conclusion, developing more sophisticated aggregation functions in window confidence might yields better performance. Moreover, employing some other rule learning algorithms other than random walk sampling might obtain more high quality rules.

## Acknowledgements

This work is supported by the National Science Foundation of China (Grant No. 62176026) and Beijing Natural Science Foundation (M22009).

## Ethics Statement

In this paper, we propose TR-Rules, a rule-based TKG extrapolation model. Compared with embedding-based models, our model has better interpretability since it can generate human-readable rules. Besides, the inference process of TR-Rules are visible, which means that users can see the whole progress of how the model gives answers including which rules are utilized and how the scores of candidates are aggregated. With these superiorities, TR-Rules can be applied into downstream applications which require high interpretabilities and reliabilities. Thus, our work complies with the ethnics policy.

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
