# OpenReview forum: "TR-Rules: Rule-based Model for Link Forecasting on Temporal Knowledge Graph Considering Temporal Redundancy"
_EMNLP/2023/Conference — EMNLP 2023 Findings_

### Official Review · Reviewer_FUMh · 2023-08-05

**Soundness:** 4

**Excitement:**

3: Ambivalent: It has merits (e.g., it reports state-of-the-art results, the idea is nice), but there are key weaknesses (e.g., it describes incremental work), and it can significantly benefit from another round of revision. However, I won't object to accepting it if my co-reviewers champion it.

**Paper Topic And Main Contributions:**

This manuscript investigates the problem of link prediction in temporal knowledge graphs. The authors propose a simple and effective method to address the limitation of temporal redundancy in rule-based methods.

**Reasons To Accept:**

The paper is well-written, and the experiments are well-conducted. The work also has good reproducibility as the authors provide the code. The results analysis is also interesting.

**Reasons To Reject:**

Firstly, the generalization ability of the proposed method is unclear. It would be interesting to know if the method could be used in other rule-based or learning-based methods. As the proposed method is an extension to TLogic, it seems a bit incremental. Providing more insights into the generalization ability of the proposed method could strengthen the paper.

Secondly, the improvements over TLogic in Table 2 are not very significant. Conducting a student t-test on multiple runs could help verify the significance of the results. This could provide readers with a better understanding of the effectiveness of the proposed method compared to TLogic.

**Reproducibility:**

5: Could easily reproduce the results.

**Reviewer Confidence:**

3: Pretty sure, but there's a chance I missed something. Although I have a good feel for this area in general, I did not carefully check the paper's details, e.g., the math, experimental design, or novelty.

---

> ### Author Rebuttal · Authors · 2023-08-29
>
> Thank you for your insightful feedbacks on our paper. We appreciate your crucial suggestions and the recognition of proposed window confidence for solving temporal redundancy.
>
> However, probably because of the organization of this paper, we think the contribution of successfully using acyclic rules is slightly underestimated. Here we would like to reclaim and draw your attention to the contribution of unprecedentedly using acyclic rules on TKG.
> Also, in terms of this, the proposed window confidence can be viewed as an exploration of how to use acyclic rules properly under temporal settings.
>
> Here, we give an intuitive example again to illustrate the necessity and advantages of using acyclic rules on TKG. In fact, sometimes acyclic rules can have equivalent cyclic rules as long as there are plenty knowledge lurking in TKGs, however it is really costly and not feasible to build TKGs to include “plenty” of knowledge in all aspect. Take a case from life for example:
>
> Acyclic rule:
>
> (X, Job, Teacher, T1) ← (X, Major in, Education, T2)
>
> Can be considered equal to cyclic rules:
>
> (X, Job, Y, T1) ← (X, Major in, Z, T2) and (Z, Is Required, Y, T3)
>
> However, for a TKG that records people's activities, it probably does not contain relation “is required” or facts like (Education major, is Required, Y, Tx). As a result, equivalent cyclic rules can not be obtained under this case. The acyclic rule itself implicitly contains the knowledge about “Education major is required for Teacher”.
> Thus, In most cases, the building cost and incompleteness of TKGs makes the utilization of acyclic rules irreplaceable.
>
> Thus, intuitively the introduction of acyclic rules should make improvements since they serve as another form of knowledge. But no one has ever managed to use acyclic rules properly for TKG extrapolation. Perhaps, as shown in ablation study, simply using acyclic rules and cyclic rules together with standard confidence gives worse performance. In this paper, we discover an effective way to utilize these type of rules containing important knowledge. We argue that acyclic rules suffer more from Temporal Redundancy. Simply combining acyclic rules with cyclic rules “disturbs” the inference of cyclic rules rather than contributes to more effective inference. The proposed window confidence alleviates this issue and makes them work together. Hence, we can say that if a model intends to capture knowledge in the acyclic form under temporal settings, the window confidence is an effective solutions.
>
> Besides, we also discover that, due to its higher sensitivity to the number of sampled walks, acyclic rule gives promising performance under temporal settings, which has not been discussed before even on static KGs.
>
>
>
> Reply for the first Question:
>
> As we've mentioned above, we think the proper utilization of acyclic rules on TKG extrapolation is one of the most important points making TR-Rules not an incremental work to TLogic. As shown in ablation study, simply using acyclic rules and cyclic rules together with standard confidence gives worse performance, which perhaps is the reason why no one has ever managed to use acyclic rules under temporal settings. However, acyclic rules indeed contain irreplaceable information for extrapolation. From this point of view, we can say we are  the first to discover how to use acyclic rules properly. The proposed window confidence makes it successful by solving temporal redundancy which is a key factor preventing acyclic rules from making sense under temporal settings. Meanwhile, window confidence also benefits cyclic rules as well.
> We genuinely appreciate that you mentioned the issue of generalization which is not included during writing and it really opens our mind to evaluate the contribution of TR-Rules from a new perspective.
> As for the generalization ability, the window confidence proposed by TR-Rules can be applied to any rule-based method as long as it involves confidence calculation under temporal settings. The confidence calculation is independent from the rule-mining stage and the rule applying stage. No matter what novel rule mining strategies or effective aggregation methods for rule application the models introduce, window confidence can yield more appropriate confidence than standard confidence. Thank you again for providing us with a new view of evaluation!
>
> Reply for the second Question:
>
> I totally understand your concern. The followings are the table of multiple runs results and the distribution of rules mined by TLogic and TR-Rules which verifies the stability and demonstrates the effectiveness of TR-Rules since the overall confidence inclines to be lower as original rules with implausible high confidence are now assigned with more accurate confidence which means rules are refined. I think this can prove the effectiveness of solving temporal redundancy. Meanwhile, thanks to the window confidence, acyclic rules can provide appropriate information which makes positive contributions to inference.
>
> ICEWS14 All Rules:
> |Models| 0-0.1 | 0.1-0.2 | 0.2-0.3 | 0.3-0.4 | 0.4-0.5 | 0.5-0.6 | 0.6-0.7 | 0.7-0.8 | 0.8-0.9 | 0.9-1.0 |
> | --- | --- | --- | --- | --- | --- | --- |--- | --- | --- | --- |
> |TLogic|0.212|0.172|0.162|0.130|0.155|0.079|0.055|0.025|0.010|0.001|
> |TR-Rules|0.427|0.214|0.153|0.083|0.078 |0.024|0.013|0.006|0.002|0.001|
>
> ICEWS0515 All Rules:
>
> |Models| 0-0.1 | 0.1-0.2 | 0.2-0.3 | 0.3-0.4 | 0.4-0.5 | 0.5-0.6 | 0.6-0.7 | 0.7-0.8 | 0.8-0.9 | 0.9-1.0 |
> | --- | --- | --- | --- | --- | --- | --- |--- | --- | --- | --- |
> |TLogic|0.312|0.173|0.130|0.097|0.097|0.064|0.053|0.039|0.026|0.009|
> |TR-Rules|0.570|0.176|0.099|0.056|0.048|0.021|0.013| 0.008|0.005|0.002|
>
> The above table shows that on both datasets, in comparison, TR-Rules gives lower but more practical confidence under temporal settings, since confidence generated by TR-Rules locals more in lower confidence section. This results complies to the intuition that by solving temporal redundancy the original inflated confidence is fixed and the temporal rules are more solid for inferencing. It demonstrates that our method indeed works and verifies the correctness of our motivations.
>
>
> To verify our claim which is “acyclic rules suffer more from temporal redundancy”, we also scrutinize the difference between distribution of the confidence score of ONLY acyclic rules. From the table below we can see the dramatical change in distribution of Acyclic rules compared with that of all rules (table above) .
>
> ICEWS14 ONLY Acyclic:
> |Models| 0-0.1 | 0.1-0.2 | 0.2-0.3 | 0.3-0.4 | 0.4-0.5 | 0.5-0.6 | 0.6-0.7 | 0.7-0.8 | 0.8-0.9 | 0.9-1.0 |
> | --- | --- | --- | --- | --- | --- | --- |--- | --- | --- | --- |
> |TLogic|0.159|0.172|0.171|0.141|0.170|0.087|0.060|0.027|0.010|0.001|
> |TR-Rules|0.361|0.226|0.171|0.095|0.092|0.028|0.016|0.008|0.002|0.0001|
>
>
> ICEWS0515 ONLY Acyclic:
> |Models| 0-0.1 | 0.1-0.2 | 0.2-0.3 | 0.3-0.4 | 0.4-0.5 | 0.5-0.6 | 0.6-0.7 | 0.7-0.8 | 0.8-0.9 | 0.9-1.0 |
> | --- | --- | --- | --- | --- | --- | --- |--- | --- | --- | --- |
> |TLogic|0.215|0.168|0.143|0.113|0.118|0.079|0.068|0.050|.034|0.012|
> |TR-Rules|0.520|0.181|0.108|0.064|0.059|0.026|0.017|0.011|0.009|0.004|
>
> Max Differences of MRR (in percent):
> |Datasets| ICEWS14 | ICEWS0515 | ICEWS18 |
> | --- | --- | --- | --- |
> |Results|43.32|47.64|30.41|
> |Max Differences|-0.11|+0.13|-0.09|
>
> The stablity of rule-based methods has been shown in TLogic (standard deviation of results is 0.0012). This condition also coincides with TR-Rules's.
>
> With extra experiments above, we can say the results of TR-Rules are not accidental and the proposed window confidence calculation indeed manages to solve temporal redundancy.
>
> We kindly request you to raise your ratings if you are satisfied with the illustrations and experiments that we made. A higher rating is an encouragement and affirmation of our work in this field. Thank you again for your thorough as well as insightful feedbacks. If you still have any concerns, please feel free to comment. Thank you!

---

### Official Review · Reviewer_cXsZ · 2023-08-11

**Soundness:** 3

**Excitement:**

2: Mediocre: This paper makes marginal contributions (vs non-contemporaneous work), so I would rather not see it in the conference.

**Paper Topic And Main Contributions:**

This paper studies the link forecasting methods on temporal knowledge graphs. They propose a window confidence algorithm to solve the temporal redundancy problem of rule-based TKG forecasting methods and simultaneously model the cyclic and acyclic rules.

**Questions For The Authors:**

A.Is the acyclic rules first proposed by this paper? if not, there should have references and more analysis about how the acyclic rules were used in previous methods. \
B. Figure 3 is confusing. The green line is not explained. The result of the blue line is not consistent with the number in Table 2. \
C. In Table 2, TR-Rules (C+A) are equivalent to TLogic with acyclic rules. However, its performance is bad than TLogic, which is weird and needs more analysis. \
D. Section 5.4 is confusing. Line 536, ‘the performance of acyclic rules improves dramatically as the number of sampled walked grows’, how does it convey? Where is the acyclic rule in Figure 3?


**Reasons To Accept:**

A. This paper investigates an interesting problem of TKG forecasting which has been neglected by previous works. \
B. The proposed method is easy to understand and a lot of effort has been done in the figures to improve the readability of the paper. \
C. The code is provided.


**Reasons To Reject:**

A. The contribution of this paper is incremental. The proposed strategy is more likely to be a trick or implementation skill in calculating confidence scores. \
B. There should be more experimental studies to verify the motivations, such as the distribution of confidence scores obtained by TLogic and TR-Rules. \
C. There are many errors in Section 5, where careful further polishing is required.


**Reproducibility:**

3: Could reproduce the results with some difficulty. The settings of parameters are underspecified or subjectively determined; the training/evaluation data are not widely available.

**Reviewer Confidence:**

4: Quite sure. I tried to check the important points carefully. It's unlikely, though conceivable, that I missed something that should affect my ratings.

---

> ### Author Rebuttal · Authors · 2023-08-29
>
> Thank you for your insightful feedback on our paper. We appreciate your dedication to ensuring the quality and clarity of our work and feel grateful that you recognize the contribution of proposed methods.
>
> However, probably because of the organization of this paper, we think that the contribution of unprecedentedly leveraging acyclic rules on TKG extrapolation task is slightly underestimated. Also, in terms of this, the proposed window confidence can be viewed as an exploration of how to use acyclic rules properly under temporal settings.
>
> Here, we give an intuitive example to illustrate the necessity and benefits of using acyclic rules on TKG. In fact, acyclic rules can be replaced by cyclic rules as long as there are plenty knowledge lurking in TKGs, however it is really costly and not feasible to build TKGs to include “plenty” of knowledge. Take a case from ICEWS14 for example:
>
> Acyclic rule:
>
> (X, Reduce Relation, South Korea, T1) ← (X, Express intent to ease sanction, North Korea, T2)
>
> Can be considered equal to cyclic rules:
>
> (X, Reduce Relation, Y, T1) ← (X, Express intent to ease sanction, Z, T2) and (Z, Is Hostile To, Y, T3)
>
> However, ICEWS14 does not contain relation “is hostile to” or any relations explicitly describing status between countries is already broken (relation “Reduce Relation” serves like an action rather than a description of already in bad status), since it is a dataset recording diplomatic events and activity. As a result, equivalent cyclic rules can not be obtained under this case. However, the acyclic rule itself implicitly contains the knowledge about “North Korea is in a bad relation with South Korea”
>
> Thus, intuitively the introduction of acyclic rules should make improvements since they serve as another form of knowledge. But as you mentioned in question C, the  results do not coincide with expectations. This is the main factor that motivates us discovering the existence of Temporal Redundancy. Because, as mentioned in section 5.3, paragraph 2, without the cyclic restriction and confined to length of 1, the matched body instances of acyclic rules repeat more in a time window and some of them might be less relevant to the head instances. In other words, acyclic rules suffer more from Temporal Redundancy. Simply combining acyclic rules with cyclic rules “disturbs” the inference of cyclic rules rather than contributes to more effective inference. Because under temporal settings, the standard confidence calculation cannot correctly evaluates rules' plausibility especially acyclic. The proposed window confidence alleviates this issue and makes them work together. Hence, we can say that if a model intends to capture knowledge in the acyclic form under temporal settings, the window confidence is an effective solutions.
> Besides, we also discover that, due to its higher sensitivity to the number of sampled walks, acyclic rule gives promising performance under temporal settings, which has not been discussed before even on static KGs.
>
> As for the reason to reject B, following your insightful suggestions, we generate the distribution of confidence scores of TLogic and TR-Rules on two datasets. The results are as follows: (numbers denote the proportion )
>
> ICEWS14 All Rules:
> |Models| 0-0.1 | 0.1-0.2 | 0.2-0.3 | 0.3-0.4 | 0.4-0.5 | 0.5-0.6 | 0.6-0.7 | 0.7-0.8 | 0.8-0.9 | 0.9-1.0 |
> | --- | --- | --- | --- | --- | --- | --- |--- | --- | --- | --- |
> |TLogic|0.212|0.172|0.162|0.130|0.155|0.079|0.055|0.025|0.010|0.001|
> |TR-Rules|0.427|0.214|0.153|0.083|0.078 |0.024|0.013|0.006|0.002|0.001|
>
> ICEWS0515 All Rules:
>
> |Models| 0-0.1 | 0.1-0.2 | 0.2-0.3 | 0.3-0.4 | 0.4-0.5 | 0.5-0.6 | 0.6-0.7 | 0.7-0.8 | 0.8-0.9 | 0.9-1.0 |
> | --- | --- | --- | --- | --- | --- | --- |--- | --- | --- | --- |
> |TLogic|0.312|0.173|0.130|0.097|0.097|0.064|0.053|0.039|0.026|0.009|
> |TR-Rules|0.570|0.176|0.099|0.056|0.048|0.021|0.013| 0.008|0.005|0.002|
>
> The above table shows that on both datasets, in comparison, confidence generated by TR-Rules incline to be in  lower confidence section. This phenomenon coincides with the intuition that by solving temporal redundancy, TR-Rules gives lower but more practical confidence under temporal settings, which can be viewed as refinement for temporal rules and the original inflated confidence is corrected. It demonstrates that our method indeed works and verifies the correctness of our motivations.
>
> We also investigate the difference between distribution of the confidence score of ONLY acyclic rules. From the table below we can see that, the difference of acyclic rules is quite larger than the difference of all rules shown above. This is a solid support for our claim which is “acyclic rules suffer more from temporal redundancy”.
>
> ICEWS14 ONLY Acyclic:
> |Models| 0-0.1 | 0.1-0.2 | 0.2-0.3 | 0.3-0.4 | 0.4-0.5 | 0.5-0.6 | 0.6-0.7 | 0.7-0.8 | 0.8-0.9 | 0.9-1.0 |
> | --- | --- | --- | --- | --- | --- | --- |--- | --- | --- | --- |
> |TLogic|0.159|0.172|0.171|0.141|0.170|0.087|0.060|0.027|0.010|0.001|
> |TR-Rules|0.361|0.226|0.171|0.095|0.092|0.028|0.016|0.008|0.002|0.0001|
>
>
> ICEWS0515 ONLY Acyclic:
> |Models| 0-0.1 | 0.1-0.2 | 0.2-0.3 | 0.3-0.4 | 0.4-0.5 | 0.5-0.6 | 0.6-0.7 | 0.7-0.8 | 0.8-0.9 | 0.9-1.0 |
> | --- | --- | --- | --- | --- | --- | --- |--- | --- | --- | --- |
> |TLogic|0.215|0.168|0.143|0.113|0.118|0.079|0.068|0.050|.034|0.012|
> |TR-Rules|0.520|0.181|0.108|0.064|0.059|0.026|0.017|0.011|0.009|0.004|
>
>
> Reply to Question A:
>
> We are the first to explore using Acyclic rules on TKG extrapolation, which we think is one of the factors that  makes TR-Rules not an incremental work to TLogic.  As you mentioned in question C, experimental results show that simply using acyclic rules and cyclic rules together with standard confidence does not give better performance under temporal settings, which probably is the reason why no one has ever attempted to utilize acyclic rules for TKG extrapolation. Our proposed method enables these two types of rules to work together and give better performance under temporal settings.
>
> Reply to Question B:
>
> I'm sorry about the confusion and really appreciate the time that you committed  into checking the experiments which helps us polish this section. We will the make the caption and illustration more clear. Here is my illustration of figure3 for you which will be added into the full paper later:
> The main purpose of Figure 3 is to show that, compared with cyclic rules, the performance of Acyclic rules is much more sensitive to the number of sampled walks during rule learning stage. This conclusion means that acyclic rules can give promising performance and can be more effective on larger TKGs.
>
> The x axis denotes the number of sampled walks during rule learning stage.
>
> The orange line represents the performance of ONLY acyclic rules with standard confidence as the number of sampled walks grows;
>
> The blue line represents the performance of ONLY acyclic rules with window confidence as the number of sampled walks grows;
>
> The green line represents the performance of ONLY cyclic rules of length 1 with standard confidence as the number of sampled walks grows.
>
> With these illustrations, the conclusions are supported more sufficiently. Conclusions including:
>
> i. the performance of acyclic rules improves dramatically as the number of sampled walks grows.
>
> ii. Compared with cyclic rules, acyclic rules are more sensitive to the number of sampled walks which yields promising performance.
>
> iii. The impact of Temporal Redundancy also intensifies as the number of walks grows since the difference between the orange line and the blue line increases .
>
> More relevant analysis can be found in line 541 to line 547.
>
> Reply to Question C:
>
> Indeed, the utilization of acyclic rules should gain improvements intuitively. However, as we claimed in section 5.3, paragraph 2, acyclic rules suffer more from temporal redundancy and the table of distribution provided for Reason to reject 2 displays a more significant change in the distribution of acyclic rules. The confidence of rules are generally overestimated by using standard confidence under temporal settings. As a result, the inference of cyclic rules is influenced by the more affected acyclic rules.
>
>
> Reply to Question D:
>
> I really appreciate the time that you committed to thoroughly check experiments . I think the explanations provided for question B can answer these questions and help you follow Section 5.4 better. If you still have any lingering doubts, please comment on this rebuttal. Thank you.
>
> We kindly request that you increase your rating to acceptance if you are satisfied with the illustrations and clarification we have made . A higher rating is an encouragement and affirmation of our work in this field. Thank you again for your insightful feedbacks which really help us to polish this paper. If you still have any concerns, please feel free to comment.

---

### Official Review · Reviewer_uQVS · 2023-08-12

**Typos Grammar Style And Presentation Improvements:** 1. In line 370, the word "learning" s…
**Soundness:** 3

**Excitement:**

3: Ambivalent: It has merits (e.g., it reports state-of-the-art results, the idea is nice), but there are key weaknesses (e.g., it describes incremental work), and it can significantly benefit from another round of revision. However, I won't object to accepting it if my co-reviewers champion it.

**Missing References:**

N/A

**Paper Topic And Main Contributions:**

he topic of this paper is closely related to the temporal knowledge graph (TKG), which has been proven to be an effective method for modeling dynamic facts in the real world. Many efforts have been dedicated to predicting future events, known as extrapolation, on TKGs. Recently, rule-based knowledge graph completion methods, considered more interpretable than embedding-based approaches, have been extended to address temporal knowledge graph extrapolation. However, in dynamic scenarios, rule-based models suffer from temporal redundancy, leading to inaccurate rule confidence calculations. As a response, the authors define the temporal redundancy problem and propose TR-Rules, a model that effectively resolves this issue through a straightforward yet powerful strategy. Moreover, to harness more latent information within TKGs, TR-Rules explores acyclic rules via temporal random walks, a facet unexplored by existing models. Experimental results on three benchmarks demonstrate that TR-Rules achieve state-of-the-art performance. Ablation studies highlight the impact of temporal redundancy and underscore the greater potential of acyclic rules, as their performance is more sensitive to the number of sampled walks during the learning phase.

**Questions For The Authors:**

A. Could the authors provide the rationale behind the selection of these baselines?
B. Have the authors considered conducting multiple experiments and averaging the results?
C. Have the authors considered releasing the paper’s code publicly?

**Reasons To Accept:**

1. This paper proposes a new method which is TR-Rules, a rule-based model for extrapolating temporal knowledge graphs. This innovative approach effectively addresses temporal redundancy through a straightforward algorithm. Notably, TR-Rules is the pioneering model to extract acyclic rules from temporal knowledge graphs.
2. This paper includes multiple baselines and conducts extensive experiments, with the obtained results sufficiently validating the scientific and rational nature of the proposed approach.
3. The primary contribution of this paper is to address Temporal Redundancy in TKG within the field of natural language, offering significant research significance.

**Reasons To Reject:**

1. Some formulas are missing labels, such as the formulas on line 171 and line 208, etc.
2. The figures in the paper are a bit blurry. Please consider replacing them with clearer ones.
3. Some sentences contain grammatical mistakes or are not complete sentences, such as, in line 370, the word "learning" should be preceded by an article such as "a" or "the".
4. The baselines in the paper lack corresponding explanations, which makes it difficult for me to understand the reasons behind their selection. Therefore, I kindly request the author to provide a brief introduction to these baselines.

**Reproducibility:**

3: Could reproduce the results with some difficulty. The settings of parameters are underspecified or subjectively determined; the training/evaluation data are not widely available.

**Reviewer Confidence:**

4: Quite sure. I tried to check the important points carefully. It's unlikely, though conceivable, that I missed something that should affect my ratings.

---

> ### Author Rebuttal · Authors · 2023-08-28
>
> We appreciate your thorough feedback on our paper. We are pleased that the reviewer recognizes the merits of our work and acknowledges the contributions we have made to the field. We would like to address the concerns and questions below.
> we are sorry about the grammatical mistakes and blurry figure and we will correct them in the final version if accepted. Thank you for your suggestions.
>
> Reply A:
> The first three models (Dismult, ComplEx, AnyBURL) are competitive link prediction models for static KG which do not consider temporal information. The reason that we include them is that we want to show TR-Rules performs much better than static models since TR-Rules captures temporal information by temporal rules sampling.
>
> (TTransE, TA-Dismult, DE-SimplE, TNTComplEx) these four models are designed for interpolation task on TKGs which are not effective in predicting future events and, instead, they focus on inferring missing facts in the past. The reason of selecting these models is that we want to show the superiority of TR-Rules over these interpolation models because TR-Rules mines temporal rules which can be considered as causality or logic between potential future events and existing facts. As a result, we can say TR-Rules is suitable for TKG extrapolation task.
>
> (RENET, CyGNet, xERTE, CENET, TLogic) these models are for extrapolation task on TKGs. We include these models to show that our model TR-Rules indeed achieves state-of-the-art performance on extrapolation on TKGs. Especially the comparison with TLogic demonstrates the effectiveness of the unprecedented use of acyclic rules and the proposed window confidence for solving temporal redundancy.
>
> Reply B:
> I completely understand your concern and here are my explanations: the main results provided in TR-Rules are the average of 5 rounds of experiments and the ablation study results are the average of 3 rounds of experiments. The following table shows the max differences of MRR in 5 rounds of experiments. Besides, stable performance is one of the property of rule-based models. As TLogic points out that running the same model with the same hyperparameter settings results in very small deviation, which coincides with conditions of TR-Rules. I'm sorry about the omissions of relevant settings and we will consider add these informations into the final version if possible. The following table displays the differences between multiple results of TR-Rules.
>
> Max Differences of MRR (in percent):
> |Datasets| ICEWS14 | ICEWS0515 | ICEWS18 |
> | --- | --- | --- | --- |
> |Results|43.32|47.64|30.41|
> |Max Difference|-0.11|+0.13|-0.09|
>
>
> Reply C:
> Yes, we have submitted our source code in the supplementary materials and we will definitely release our source code if this paper is accepted.
>
> We kindly request that you increase your rating to acceptance, especially for the soundness score, in light of the improvements and illustrations we have made. A higher rating is an encouragement and affirmation of our work in this field. Thank you again for your thorough feedbacks. If you still have any concerns, please feel free to comment.

---

### Meta-Review · Area_Chair_ayvs · 2023-09-18

**Recommendation:** 4

**Metareview:**

This paper proposes a new method which is TR-Rules, a rule-based model for extrapolating temporal knowledge graphs. This innovative approach effectively addresses temporal redundancy through a straightforward algorithm. However, the reviewers also state the technical contribution is incremental, more experimental studies are expected to support the claims.

---

### Decision · Program_Chairs · 2023-10-07

**Decision:**

Accept-Findings

**Comment:**

This paper proposes a new method which is TR-Rules, a rule-based model for extrapolating temporal knowledge graphs. This innovative approach effectively addresses temporal redundancy through a straightforward algorithm. However, the reviewers also state the technical contribution is incremental, more experimental studies are expected to support the claims.